# Incidence of Inadequate Transfer of Passive Immunity in Dairy Heifer Calves in South Australia

**DOI:** 10.3390/ani12212912

**Published:** 2022-10-24

**Authors:** Rebel Skirving, Cynthia D. K. Bottema, Richard Laven, Do T. Hue, Kiro R. Petrovski

**Affiliations:** 1Davies Livestock Research Centre, School of Animal & Veterinary Sciences, University of Adelaide, Roseworthy, SA 5371, Australia; 2Gambier Vets Pty. Ltd., Mount Gambier, SA 5290, Australia; 3College of Veterinary Science, Massey University, Palmerston North 4442, New Zealand; 4Faculty of Animal Science, Vietnam National University of Agriculture, Trau Quy, Gia Lam, Hanoi 12406, Vietnam; 5Australian Centre for Antimicrobial Resistance Ecology, School of Animal & Veterinary Sciences, University of Adelaide, Roseworthy, SA 5371, Australia

**Keywords:** colostrum, failure of passive transfer, dairy cattle, Australia, refractometer

## Abstract

**Simple Summary:**

Calves are born with an underdeveloped immune system, and therefore must absorb immune components, such as immunoglobulins, from their dam’s first milk (colostrum) to help fight infectious disease. If they do not receive enough good-quality colostrum within 24 h of birth, their immune status is compromised, and they are more likely to have poor growth and health in their first 12 weeks of life. Colostral immunoglobulin uptake can be estimated by measuring serum total protein with a digital refractometer in calves 1–7 days of age. The aim of this study was to estimate the uptake of colostral immunoglobulin in female dairy calves from five pasture-based dairies in the southeast of South Australia to evaluate the proportion of calves that did not receive an adequate amount of colostrum.

**Abstract:**

The objective of this observational study was to estimate the incidence of inadequate transfer of passive immunity (ITPI) on five pasture-based dairy farms in South Australia. Heifer calf uptake of colostrum was evaluated within the first 1–7 days of age (*n* = 2638) using a digital refractometer to estimate each calf’s serum total protein concentration, as an indicator of colostrum uptake. Results of <51 g/L indicated inadequate transfer of passive immunity (ITPI). The data showed that the incidence of ITPI on the farms was 6.5%, 31.3%, 48.8%, 49.7% and 52.4%. The incidence of ITPI was calculated in relation to the age of the calf at testing and the breed of calf, and no significant differences were found. A significant difference was found in the incidence of ITPI when comparing the calf’s first feed after separation from the dam (colostrum versus a colostrum-transition milk mixture). The farm with the lowest incidence of ITPI collected calves twice a day, measured colostrum quality on farm with a Brix refractometer and ensured that each calf received an appropriate amount of high-quality colostrum soon after collection. Further studies are required to establish the risk factors of ITPI in South Australian dairy heifers.

## 1. Introduction

Cows have a cotyledonary synepitheliochorial placenta [1,2], which prevents immunoglobulins in the dam’s blood from entering the fetal circulation and results in calves being born virtually agammaglobulinaemic. Therefore, to obtain an effective immune response against pathogens in their first few weeks of life, neonatal calves rely on the absorption of maternal immunoglobulins as well as other immune-related proteins and cells from colostrum [3,4,5]. Calves that receive adequate amounts of good-quality colostrum within the first 24 h of birth will generally have better health and growth in the pre-weaning stage compared to calves with an inadequate uptake of colostrum [6,7,8,9,10]. The concept of ‘adequate’ uptake of colostrum has historically focused on the immunological benefits of colostrum, primarily the immunoglobulin component that is transferred from colostrum into the calf’s bloodstream (mainly immunoglobulin G (IgG) in cattle [10]). The absorption of maternal IgG from colostrum through the neonatal gut results in the ‘transfer of passive immunity.’ The level of passive immunity acquired by the calf has been measured and estimated by various methods and, for each method, there has been a variety of suggested thresholds for determining whether a calf has received ‘adequate’ immunity (Table 1). Serum has been the most common sampled tissue, usually in calves older than 24 h of age and less than seven days of age. Direct measures of IgG are generally considered to be the most accurate for estimating the transfer of passive immunity, but these tests are expensive, slow and only measure IgG. Alternate tests that indirectly estimate the uptake of IgG or other colostral components have also been used. For example, measuring serum total protein with a refractometer is a quick, cheap and reliable way to estimate the uptake of IgG, since most of the serum protein in young calves is IgG [11].

There has been no agreement on the ‘correct’ threshold to define ITPI, because the ‘ideal’ threshold will depend on the outcome being measured and the management system used to raise calves. For instance, Windeyer et al. (2014) [26] found that a serum total protein concentration threshold of 55 g/L was most predictive of diarrhea in calves less than five weeks of age (87% negative predictive power), whereas a threshold of 52 g/L was most predictive of death before five weeks of age (98% negative predictive power) [26].

The failure to reach desired levels of immunity, regardless of the measure used and the nominated threshold, has been referred to by a number of different terms, including ‘failure of passive immune transfer’ [29], ‘failure of passive immunoglobulin transfer’ [30] and ‘failure of passive transfer’ [31,32]. These terms are inaccurate for two reasons. Firstly, they imply that the transfer of immunity is a passive process. The absorption of immunoglobulins and other immune components from colostrum across the gut wall is an *active* process, which results in *passive* immunity [17]. Secondly, terms using the word ‘failure’ imply that there has been *no* transfer of passive immunity, whereas, in most cases, there has been some transfer of passive immunity, but just not a sufficient amount to provide satisfactory protection against morbidity and/or mortality. Other authors have addressed these inaccuracies by referring to the occurrence as ‘inadequate transfer of colostral immunoglobulins’ [33] or ‘inadequate transfer of passive immunity’ [34]. The later term is favored herein, as it recognizes that there are several immunologically active substances absorbed from colostrum important to confer passive immunity to calves, such as fats, minerals and leucocytes, in addition to immunoglobulins [4].

In a recent review, Lombard et al. (2020) [17] evaluated several different thresholds used to define inadequate transfer of passive immunity in calves. From this review, a grading system, based on protection against morbidity in the US system, was developed to define four levels of transfer of passive immunity: excellent, good, fair and poor. Each of the classification levels identified thresholds for serum IgG, equivalent serum total protein and equivalent Brix %.

Calves with values in the ‘poor’ category are considered to have ITPI and, correspondingly, calves enrolled herein with <51 g/L serum total protein concentration were classified as having ITPI. Lombard et al. (2020) [17] recommended that the standard for transfer of passive immunity on American dairy farms should be >40% calves in the excellent group, 30% calves in the good group, 20% calves in the fair group and <10% calves in the poor (ITPI) group, a standard which should be achievable in Australian conditions.

Given the clear advantages of good colostrum uptake for dairy calves, it is surprising to find that many dairy regions of the world still have high rates of ITPI. Studies in Canada have estimated the incidence of ITPI to be 25–37% [35,36] and an estimated incidence of ITPI in America of 13% has been reported [37]. Even higher levels of ITPI have been reported in New Zealand, ranging from 33% to 45% [32,38,39]. Australian studies are few, but of those that have been conducted, estimated rates of ITPI ranged between 8.7% and 41.9% [40,41,42,43,44]. Approximately 5% (69,000 head) of Australia’s dairy cattle are in South Australia [45] and, to the best of our knowledge, there no studies have been conducted to assess the incidence of ITPI in this state.

The aim of this study was to estimate the incidence of ITPI in dairy-heifer calves on five pasture-based dairy farms in the southeast of South Australia under different management systems to assess the risk factors for ITPI and provide data for benchmarking and comparisons with other dairying areas of Australia and internationally.

## 2. Materials and Methods

### 2.1. Study Farms

A convenience sample of five pasture-based dairy farms from the southeast of South Australia were recruited for this study, based on their proximity to Mount Gambier, accuracy of record-keeping and different calf management styles. The breed of tested cattle was Holstein, Jersey or crossbred (Table 2).

Calves from all five farms were housed in open sided sheds, separated from their dams. The manager of Farm 4 measured colostrum quality on farm with a Brix refractometer and only used colostrum >22% Brix for the calves’ first feed. Managers on the other four farms used colostrum of unmeasured quality. Herein, colostrum was defined as the mammary secretion collected from a cow only during the first milking postpartum. Any mammary secretion collected from the cow at subsequent milkings, through to and including four days postpartum, was defined as transition milk.

### 2.2. Sample Collection

Visibly sick or dehydrated calves at the time of sampling were not included in the study. From each of the farms, all heifer calves born during a specified calving season were included in the study and sampled between 1–7 days of age. Calves from four farms (Farms 2–5) were sampled from July 2020 to May 2021, and calves from the other farm (Farm 1) were sampled from Feb 2018 to Oct 2020. Each enrolled heifer calf was identified by ear tag and had 6–8 mL whole blood collected from the jugular vein into a sterile 9 mL plain vacutainer (Becton Dickinson, Plymouth, UK). Blood samples were left to clot for 2–24 h in a refrigerator (4 °C) before being centrifuged for 10 minutes at approximately 2000× *g,* using a Premiere Low Speed Centrifuge (model XC-2415, C&A Scientific, Sterling, VA, USA). After centrifugation, serum was immediately tested for total serum protein concentration using a digital refractometer, and the remainder of the serum was stored at −20 °C, then transported to the University of Adelaide for storage at −80 °C and further testing.

### 2.3. Refractometry for Total Soluble Protein Concentration (TP-R)

All serum samples were tested for total soluble protein concentration (TP-R) using a handheld Atago digital refractometer (Saitama, Japan), with a protein measurement range of 0–120 g/L. The refractometer was calibrated with distilled water between each sample, and each sample was tested in duplicate.

### 2.4. Bovine IgG ELISA Assay

To validate the accuracy of refractometry results, a portion of the frozen serum samples (*n* = 132) were selected at random for bovine IgG ELISA testing, following procedures described by Hue et al. (2021) [46]. The total soluble protein concentration was then compared to the ELISA IgG results, using the categories from the review by Lombard et al. (2020) [17].

### 2.5. Statistical Analysis

#### 2.5.1. Data Manipulation

Following sample collection, each calf’s date of birth was cross-checked and samples from any calves less than 24 h old or more than 7 days old were rejected. The remaining samples were grouped according to the refractometry serum total protein or serum IgG concentrations. The serum total protein categories were defined as excellent, good, fair or poor (≥62, 58–61, 51–57 and <51 g/L, respectively), where a ‘poor’ result equated to inadequate transfer of passive immunity [17]. In addition, the serum IgG concentrations were defined as excellent, good, fair or poor (≥25.0, 18.0–24.9, 10.0–17.9 and <10 g/L, respectively), where a ‘poor’ result equated to inadequate transfer of passive immunity [17]. The agreement between categories of serum protein and serum IgG were classified as ‘agreed’ (both in the same category, e.g., serum total protein and IgG concentration were in the ‘poor’ category) or ‘disagreed’ (when categories for a sample did not match, e.g., serum total protein in the ‘poor’ category but IgG concentration was in the ‘fair’ category).

#### 2.5.2. Data Analyses

Data were analyzed using SAS version 9.4 (Statistical Analysis Software, Cary Inc.) statistical package. The correlation between the refractometry and IgG levels measured by ELISA was estimated using the PROC CORR. Correlations were interpreted as weak (0.0–0.3), moderate (0.31–0.69), strong (0.70–0.99) or exact (1.0) [47].

The agreement between categories of serum protein and serum IgG was analyzed by a kappa test in PROC FREQ. Kappa agreement results were interpreted as values ≤0 as indicating no agreement, 0.01–0.20 as none to slight, 0.21–0.40 as fair, 0.41–0.60 as moderate, 0.61–0.80 as substantial, and 0.81–1.00 as almost perfect agreement [48].

The effect of calf age (days), breed, farm and type of colostrum at first feed, on the odds of ITPI (<51 g/L) was estimated using logistic regression (PROC GLIMMIX) with frequency of collection after using calving as a covariate.

## 3. Results

The correlation between serum total protein readings (TP-R) and bovine ELISA IgG in 132 randomly selected samples was strong at 0.85 (95% CI 0.80–0.89). The agreement between categories of serum protein and serum IgG, based on Lombard’s categories [17], was substantial at 0.76 (95% CI 0.69–0.83) (Table 3).

The number of heifer calves tested varied between 139 and 1958 per farm. The incidence of ITPI varied between farms from 6.5% to 52.4% (Table 4).

The proportion of calves with ITPI was significantly lower (12.9%) for calves fed colostrum at the first feeding, compared with calves fed a mixture of colostrum and transition milk (*p* = 0.015). There was no significant association between the proportion of calves with ITPI and either the age of calf at testing or the breed of calf (Table 5).

## 4. Discussion

The main aim of this observational study was to estimate the incidence of inadequate transfer of passive immunity (ITPI) on five pasture-based dairy farms in South Australia, in order to set a benchmark for comparison to other dairying regions nationally and internationally, and to compare to the recommended standards [17]. The incidence of ITPI on the participating farms ranged from 6.5 to 52.4%. Risk factors for ITPI were also assessed, and, in this study, it was found there were no significant differences in risk associated with age of calf at time of testing, breed of calf, or frequency of collection. The type of colostrum used for the calf’s first feed after separation from the dam did have an effect on the risk of calves having ITPI.

Serum refractometry was used in this study to estimate the uptake of colostrum due to the practical application of refractometers in field conditions. While other methods of assessing colostrum uptake may be considered superior (e.g., radial immunodiffusion, ELISA), their practicality in field conditions is limited by the extended time taken to return results and the high cost per test. In contrast, serum refractometry is a relatively cheap and quick method to estimate colostrum uptake, and has good correlations with other measurement methods, such as ELISA. In this study, the correlation between TP-R and bovine ELISA IgG was strong (0.85). A study by Cuttance et al. (2017) assessed serum IgG levels against three indirect measures of IgG (serum TP concentration, Brix % and gamma-glutamyl transferase activity) and data from that study showed that serum total protein concentration had the highest correlation with serum IgG (95% accuracy), with an optimized threshold of 52 g/L. This indicates that serum TP can be used to estimate IgG levels, and therefore to estimate the transfer of passive immunity to the calf. Of the 132 samples tested, the agreement between TP-R and IgG was substantial (0.76) with no tendency of under- or over-estimating the IgG by measuring the TP-R.

To estimate the colostrum uptake based on serum refractometer readings, it was necessary to omit any calves showing signs of dehydration, since dehydration can hemoconcentrate the serum and the total protein readings for these calves will be artefactually increased [41]. Although the number of such calves were small in relation to the sample size (<2% total), it is possible that their omission may have altered the true proportion of calves in each category. However, most calves in this study were healthy and well hydrated at the time of sampling, and therefore should represent a reliable sample of the population studied. When using serum refractometry on-farm to assess levels of ITPI, particularly with a small sample size, it is important to document the number of sick or dehydrated animals not sampled, as their omission may affect results.

For the farms participating in the study, the percentage of calves classified as having ITPI ranged from 6.5 to 52.4%. This large difference was likely a result of different management practices on each farm. On the farm with the lowest incidence of ITPI, calves were collected from the calving paddock within 12 h of birth, and all calves were given two feeds of 2–3 L of colostrum only (not transition milk) within 12 h of arriving in the calving shed. Colostrum quality was measured by BRIX refractometer on farm and only good-quality colostrum was used to feed calves (BRIX > 22%). On the farm with the highest incidence of ITPI, the calves were collected from the calving paddock once a day. For the first colostrum feed after collection on this farm, the calves were offered a maximum of 2 L of colostrum of untested quality, but some calves did not drink the full 2 L, possibly because they had suckled from their dams. Cuttance et al. (2022) found approximately 58% of dairy calves left with their dams will suckle in the first six hours after birth, but the amount of colostrum ingested can vary [32,49]. The high proportion of calves with ITPI on this farm could be the result of calves being older on arrival to the shed (up to 24 h old) and some calves receiving less than 2 L of colostrum of untested quality. Between the five farms, there was no increased risk of ITPI based on calves being collected from the calving paddock once a day versus twice a day. However, it is possible that calf collection frequency, in combination with other management factors (such as volume and quality of colostrum at first feed) may have a combined effect in the incidence of ITPI.

Most calves from each farm were categorized in either the ‘excellent’ or ‘poor’ classification for transfer of passive immunity, presumably as a result of the narrow range of readings applicable to the ‘good’ and ‘fair’ categories. For example, the reference range for an ‘excellent’ result was a refractometer measurement equal to or over 62 g/L (sample range 62–100 g/L), whereas the reference range for a ‘good’ result was a measurement specifically between 58 and 61 g/L. The notable exception to this observation was Farm 4, where the vast majority of calves were in the ‘excellent’ and ‘good’ ranges. The calf and colostrum management on this farm were exceptional, with calves diligently collected from the calving paddock twice a day, housed in a clean, spacious shed, monitored frequently and provided with a large quantity of high-quality colostrum within 12 h of birth. The incidence of ITPI on Farm 4 was 6.5%, which is similar to the average of 8.7% from a recent study of 26 dairy farms in a temperate region of Western Australia [43]. The manager of Farm 4 was able to achieve the standard suggested by Lombard (2020), with >40% calves in the excellent category, and <10% in the poor category. In contrast, the incidence of ITPI on the other four farms (31.3%–52.4%) was much higher than the recommended standard, but similar to other Australian and New Zealand studies, which showed ITPI results between 33% and 45% [32,39,40].

The results of the study herein show there can be a large variation in incidence of ITPI between farms in the same geographical region. The data also indicate that colostrum and stock management techniques have the potential to have a marked effect on ITPI, regardless of the size of the farm or the heifer breed. Although there were apparent differences in the proportion of calves with ITPI based on their age (in days), breed and frequency of collection, these differences were not significant. The effect on ITPI may be more substantial if these factors are considered in combination, rather than individually. It should be also noted that the seasonal effect of colostrum quality was not specifically investigated in this study and remains to be explored. Further studies are required to establish the role of these risk factors for ITPI, both individually and in combination.

## 5. Conclusions

This study showed there is a high incidence of ITPI in South Australian dairy heifers; however, there is substantial variation between individual farms. In South Australian pasture-based farms, it is possible to achieve very high levels of passive transfer of immunity with diligent calf and colostrum management techniques; however, most farms fall well below the recommended standard.

## Figures and Tables

**Table 1 animals-12-02912-t001:** Examples of different measures and estimates of transfer of passive immunity in calves from serum samples. Values below the thresholds are indicative of inadequate transfer of passive immunity (ITPI).

Method	Measure of IgG	Calf Age (Days)	Threshold	Reference
Brix refractometry	Indirect	1–7	7.8% 8.4% 8.8%	[12] [13] [14]
ELISA IgG	Direct	1–7	10 g/L >20 g/L	[14,15,16,17] [18]
Gamma-glutamyl transferase activity	Indirect	<5	250 IU/L >100 IU/L	[14] [19]
Radial immunodiffusion IgG	Direct	1–7	10 g/L 20–25 g/L	[20,21,22] [18]
Total protein concentration by refractometry	Indirect	1–7	42 g/L 50–52 g/L 52–57 g/L 55 g/L 58–63 g/L	[23] [14,17,22,24,25] [26] [20,27] [18]
Zinc sulphate turbidity IgG	Indirect	1–7	200 mg/L 350 mg/L	[27] [28]

**Table 2 animals-12-02912-t002:** Description of dairy farm management and study population. All calves sampled at 1–7 days of age.

Farm	Number of Sampled Calves	Breed Distribution	Number of Milking Cows	Frequency of Collection from Paddock	First Feed Source
Farm 1	1958	1958 (H ^1^)	1900	Twice a day	CO ^2^ (unmeasured) + TM ^3^
Farm 2	151	14 (J ^4^) 137 (C ^5^)	410	Once a day	CO (unmeasured)
Farm 3	208	1 (H) 18 (J) 189 (C)	600	Once a day	CO (unmeasured) + TM
Farm 4	139	139 (H)	380	Twice a day	CO (>22% Brix)
Farm 5	182	14(J) 168 (H)	500	Twice a day	CO (unmeasured)

^1^ H—Holstein; ^2^ CO—colostrum (only first milking postpartum); ^3^ TM—transitional milk (mixture of second to eight milking post-partum); ^4^ J—Jersey; ^5^ C—crossbreed.

**Table 3 animals-12-02912-t003:** Comparison of 132 randomly selected samples tested for both serum total protein concentration and serum IgG concentration.

Serum Total Protein Category	Serum IgG Concentration Category				Total
	Poor	Fair	Good	Excellent	
Poor	34	10	0	0	44
Fair	5	25	1	1	32
Good	0	9	3	6	18
Excellent	1	1	4	32	38
Total	40	45	8	39	132

**Table 4 animals-12-02912-t004:** Proportion of heifer calves, in excellent, good, fair and poor categories for transfer of passive immunity [17], from five dairy farms in South Australia. The transfer of passive immunity was estimated using serum refractometry.

Farm	Variable	Adequacy of Transfer of Passive Immunity			
		Adequate			Inadequate
		Excellent ≥62 g/L	Good 58–61 g/L	Fair 51–57 g/L	Poor <51 g/L
Farm 1 ^a^ *n* = 1958	N	506	156	341	955
	%	25.8	8.0	17.4	48.8
Farm 2 ^b^ *n* = 151	N	51	4	21	75
	%	33.8	2.7	13.9	49.7
Farm 3 ^b^ *n* = 208	N	52	16	31	109
	%	25.0	7.7	14.9	52.4
Farm 4 ^a^ *n* = 139	N	77	34	19	9
	%	55.4	24.5	13.7	6.5
Farm 5 ^a^ *n* = 182	N	60	28	37	57
	%	33.0	15.4	20.3	31.3

Frequency of calf collection from calving paddock: ^a^ twice per day; ^b^ once per day.

**Table 5 animals-12-02912-t005:** Proportion of heifer calves with inadequate transfer of passive immunity (95% confidence intervals) for a variety of factors, from five dairy farms in South Australia, adjusted for the effects of farm and frequency of calf collection from calving paddock per day. The transfer of passive immunity was estimated using serum refractometry.

Effect	Description	Proportion of Calves with Inadequate Transfer of Passive Immunity	95% Confidence Interval
Age (days)	1	24.7 (299/1202)	(12.2–43.5)
	2	25.5 (204/809)	(12.7–44.6)
	3	27.1 (87/334)	(13.6–46.8)
	4	26.7 (46/171)	(13.0–47.2)
	5	37.6 (27/70)	(18.5–61.5)
	6	43.3 (17/39)	(20.5–69.4)
	7	25.8 (4/13)	(7.7–59.1)
Breed	Crossbreed	25.6 (89/326)	(10.2–51.0)
	Holstein	36.8 (846/2266)	(19.5–58.2)
	Jersey	27.5 (13/46)	(11.1–53.5)
Colostrum type	Colostrum ^1^	12.9 (63/472)	(3.5–37.3) ^#^
	Colostrum and transition milk ^2^	54.8 (1194/2166)	(31.1–76.6) ^#^

^1^ Colostrum from first milking post-partum only ^2^ Mixture of colostrum and transition milk (from 2nd–8th milkings of the lactation). ^#^ Differences within the measure at *p* = 0.015.

## Data Availability

Data available on request from R. Skirving.

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
