# Peer review of "Incidence of Inadequate Transfer of Passive Immunity in Dairy Heifer Calves in South Australia"

_animals, 2022, doi:10.3390/ani12212912_

Round 1

Reviewer 1 Report

This observational study presents TPI results from a large number of calves born in 5 commercial dairy farms. However, most of the animals were sampled from a single farm. The data set is valuable because of its size but the data analysis methods and the overall presentation of the manuscript need refinement.

The body of the manuscript needs mayor editing. Information currently presented in the results section (i.e. n of sampled calves/farm) should be presented in the M&M section. In addition, work should be devoted to have the content in each section to actually match the corresponding heading. As it is right now, the sections content is mixed. For example sections 2.1. Study farms and 2.2. Sample collection (see below for further details).

As it is right now, it is uncertain if the statistical analysis approach is correct for the data. However, it seems that, perhaps with the exception of age and breed (can’t tell certainly since distribution of age of sampled calves or breed by farm is not provided), the evaluated variables could be considered a “farm effect” (i.e. frequency of collection after calving and type of colostrum at first feed), and thus concerns of collinearity among the variables in the model and consequently model estimates and inference arise.

Mayor details on distribution of animals in the evaluated categories are needed to evaluate the adequacy of the statistical analyses presented (i.e. number of calves fed colostrum vs. transition milk). These numbers should also be presented by farm.

To this reviewer, only those “risk-factors” observed in all the evaluated herds should be assessed. Otherwise it is hard to say if it is the risk factor or the “farm effect”.

While the evaluation and comparison of the refractometer readings and the IgG determined by ELISA is commendable, further details should be provided.

The introduction should address the use of TP instead of IgG, pros and cons and what is actually measured in the TP fraction. It would also be of value to refer to the other components of colostrum (minerals, fats…) as it is becoming more and more evident that not only IgG provide an advantage to the calves (even to the immune response), it should be recognised in every paper referring to colostrum.

Some specific comments below.

Simple summary: The simple summary should reflect that only total proteins were evaluated.

L 18: Since colostrum provides much more than IgG, it seems inadequate to refer to “uptake of colostrum” when only IgG are measured. Please, change to colostrum immunoglobulin.

30: Add space “mixture).The farm”

113: What do you mean by “different calf management styles”

2.1. Study farms: Please provide further details of farms. Although it could be inferred from the breeds in the study farms, it should be clearly stated that the farms where dairy farms. Please, add this information here. Herd size and some more description of general production and management are also needed

2.1. Study farms and 2.2. Sample collection: please review these sections, some of the details of sample collection are mixed in the above section. Details of sample collection such as time relative to birth should be easily identified under the Sample collection subheading. The period of sample collection should also go under sample collection. Seasonal effects have been described for colostrum composition, please, ensure this is discussed.

138: The reason for the transport to the university of Adelaide was likely to get tested rather than to be stored, right?

143-145: I don’t think that it is expected to have exactly equal readings. If you have multiple readings/sample, then report the associated CV

143: 0 – 12 g/100mL. Please, provide the range in the same units as the reported measurements.

2.4. Bovine IgG ELISA assay: Indicate here how many samples conform “the subset” and add a brief description of the procedure (the ELISA testing and the comparison between refractometry results and this.

158: As it is a calf with a value of 25 could be in 2 different categories (≥ 25, 18 - 25, 10 - 18 and < 10 g/L,)

160: “Could” or was actually classified

175: What do you mean by “frequency of collection after calving as a covariate”? Pease, clarify.

REFERENCES

References are presented inconsistently. Some include all author and others the first author followed by et al.. Please, check instruction for authors and adjust references consistently.

Author Response

Answers in the attached file

Reviewer 2 Report

The manuscript is very well written and informative. Study was well designed and is well described. I have the following specific suggestions:

1. The introduction is long. Table 1 probably does not need to be included, it adds very little to the paper and the points from that table which are relevant to this work could be added in a single sentence line 52.

2. Line 119. It would be helpful to state here that "on the farm that measured colostrum, only good quality colostrum (Brix >22%) was selected for feeding to calves" rather than leave the reader wondering about it until they find it in the discussion section.

3. Line 189. In the table heading for Table 4 please don't refer to these as "Lombard's categories". Title could be "Proportion of heifer calves in excellent, good, fair, and poor transfer of passive immunity categories, from five dairy farms......" You could include the reference [16] in the table heading rather than having to list it multiple times at the top of each column.

4. Lines 194-196. This sentence could be modified to read "The proportion of calves with ITPI was significantly lower (12.9%) for calves fed colostrum at the first feeding, compared with calves fed a mixture of colostrum and transition milk (P = 0.015). There was no significant association between the proportion of calves with ITPI and either age of the calf at testing, or breed of calf.

Author Response

Answers in the attached file

Reviewer 3 Report

In the study evaluation of % calves with inadequate transfer of passive immunity from 5 dairy farms in Australia is presented. Although, this problem is recognized from decades all over the word, it is surprising that sill in XXI century farms are struggling with proper procedure in order to provide adequate quality and quantity of colostrum. Therefore it is necessary to control effectiveness of passive immunity transfer.

My comments concern the description of farms in M&M. Please give characteristic of each farm in the way that reader can understand which procedures and which systems were on which farm. Right now (lines 113-119) it is impossible to combine the information’s about farms. This is important, especially when in the discussion (i.e. lines 241-247) you are connecting the difference in results to different management farm procedures. The farm characterization may be done in the table.

If only 32 samples were evaluated in ELISA and then the comparison with total protein concertation was done with different levels (in category “good” only 3 samples were) the agreement may be not very accurate. If there is no possibility to evaluate more samples in ELISA, this should be mention as limitation of the study.

If you notice that the highest % of calves with excellent level of passive immunity transfer was reach on the farm where the management was exceptional, this may be also one of conclusion from your study.

Minor comments:

In table 4-Farm 1 text is bold but others don’t. Please correct it.

Author Response

Answers in the attached file

Round 2

Reviewer 1 Report

Thank you for working on addressing my previous comments.

The study has great value as a descriptive study however, I am still concerned about the statistical analysis approach. Given the breed distribution in the study (2,266 HO; 46 JE; 326 Cross), I’m unsure of its adequacy to assess this effect. Please, provide the distribution of calves (numbers) fed colostrum or TM and the distribution of calves’ age. It could be added to Table 6; numerator and denominator could accompany the %. Some of the 95% CI presented are remarkably wide, perhaps due to a small number of animals in these groups.

160-161. It would have been more “ethically sound” to check calves’ age before having them undergo the procedure of jugular blood sampling.

Please, check the manuscript for consistent presentation of numbers (2,000 vs 2000), bold font (Line 279), spaces between symbols and numbers (Lines 182 vs 166), and italics (Table 4).

Author Response

Thanks to the reviewer for the constructive criticism.

We have addressed all reviewers comments in the attached file.
